# Metasurface-driven full-space structured light for three-dimensional imaging

Gyeongtae Kim[1,7], Yeseul Kim[1,7], Jooyeong Yun [1,7], Seong-Won Moon[1], Seokwoo Kim[1], Jaekyung Kim[1], Junkyeong Park[1], Trevon Badloe [1], Inki Kim [1,2,3] ✉ & Junsuk Rho [1,4,5,6] ✉

Structured light (SL)-based depth-sensing technology illuminates the objects with an array of dots, and backscattered light is monitored to extract three-dimensional information. Conventionally, diffractive optical elements have been used to form laser dot array, however, the field-of-view (FOV) and diffraction efficiency are limited due to their micron-scale pixel size. Here, we propose a metasurface-enhanced SL-based depth-sensing platform that scatters high-density ~10 K dot array over the 180° FOV by manipulating light at subwavelength-scale. As a proof-of-concept, we place face masks one on the beam axis and the other 50° apart from axis within distance of 1 m and estimate the depth information using a stereo matching algorithm. Furthermore, we demonstrate the replication of the metasurface using the nanoparticle-embedded-resin (nano-PER) imprinting method which enables high-throughput manufacturing of the metasurfaces on any arbitrary substrates. Such a full-space diffractive metasurface may afford ultra-compact depth perception platform for face recognition and automotive robot vision applications.

Laser-based three-dimensional (3D) surface imaging technology has exhibited potential in a variety of sensor applications such as augmented/virtual reality (AR/VR), autonomous driving, robot vision, and face recognition in mobile devices with the advances in high-speed computing platforms[1]. The depth information of the 3D objects is estimated by illuminating targets with the controlled laser beam through transmitters and by monitoring backscattered light from them at the receivers. Depending on the depth calculation methods at the detectors, the 3D imaging systems are classified into time of flight (TOF) and structured light (SL) types. Using a pulsed laser to image 3D objects, the depth information is extracted from the TOF, which is the measured time delay between the illuminated pulse light and the backscattered signal. To achieve a wide field of view (FOV), the TOF must be sequentially measured at each point while rotating and scanning with the laser[2,3]. Laser scanning systems are realized by mechanically rotating mirrors, or microelectromechanical systems (MEMS). However, mechanical rotating of mirrors demands high-power due to the inertia of the bulky components, resulting in a decreased frame rate of object acquisition. MEMS can reduce the power consumption but oscillating-based operation limits the scannable scene into one-dimensional (1D) and is vulnerable to vibrations and shocks. Recently, electrical beam steering using transparent conductive oxides (TCOs)[4–6] and liquid crystals (LCs)[7,8] have been proposed, complying with the physical mechanism of the solid-state optical phased array antennas, but still suffering from small FOV.

[1]Department of Mechanical Engineering, Pohang University of Science and Technology (POSTECH), Pohang 37673, Republic of Korea. [2]Department of Biophysics, Institute of Quantum Biophysics, Sungkyunkwan University, Suwon 16419, Republic of Korea. [3]Department of Intelligent Precision Healthcare Convergence, Sungkyunkwan University, Suwon 16419, Republic of Korea. [4]Department of Chemical Engineering, Pohang University of Science and Technology (POSTECH), Pohang 37673, Republic of Korea. [5]POSCO-POSTECH-RIST Convergence Research Center for Flat Optics and Metaphotonics, Pohang 37673, Republic of Korea. [6]National Institute of Nanomaterials Technology (NINT), Pohang 37673, Republic of Korea. [7]These authors contributed equally: Gyeongtae Kim, Yeseul Kim, Jooyeong Yun. ✉e-mail: inki.kim@skku.edu; jsrho@postech.ac.kr

In contrast, SL-based 3D imaging systems use specially designed two-dimensional (2D) light patterns to project the object, efficiently enlarging the FOV without any actual scanning of light. When the surface of a 3D object is nonplanar, it distorts the projected SL pattern, and the surface profile can be calculated from distorted light pattern using a variety of algorithms[9]. Such a 2D operation can increase the frame rate of object acquisition with less computational load by facilitating the simultaneous imaging of multiple objects. Thus far, diffractive optical elements (DOEs) or spatial light modulators (SLMs) are used to generate 2D light patterns. The conventional micro-sized DOEs should be etched to multiple depths to modulate the phase with multiple steps which induce challenges in terms of fabrication. In addition, both DOEs and SLMs have large pixel sizes on the order of microns, resulting in low efficiency and uniformity of the diffracted beam arrays, especially at large angles. The uniformity of diffracted beams from SLM can be resolved by calculating phase profile of SLM using vectorial Debye approximation[10] which considers the depolarization effect when focusing light with a high numerical aperture objective[11,12]. However, the requirement of bulky objective lenses is still challenging for miniaturization of SL illuminating system. Therefore, the demands for next-generation, compact, and lightweight illumination methods of SL imaging systems are increasing, in both the academic and industrial fields, and nanophotonics is a promising candidate to overcome the challenges of current SL-based 3D imaging systems[13].

Metasurfaces, 2D arrangements of designed nanostructures named meta-atoms, dominantly inherit their properties from the composing building blocks, i.e., resonant and waveguiding effects[14,15]. They have been used to demonstrate extraordinary light modulation at the nanoscale, which cannot be observed using conventional optical devices[16–18]. With intense studies of light-matter interactions in subwavelength-scale structures[19–22], metasurfaces have demonstrated exceptional functionalities in a variety of applications such as imaging[23–25], color filters[26–28], holographic displays[29,30], polarizing elements[31,32], and beam shapers[33,34]. In terms of metasurface for 3D imaging systems, the subwavelength pitch pixels of metasurface can improve the FOV and diffraction efficiency compared to conventional DOEs and SLMs by modulating the phase at a subwavelength resolution. Using metasurfaces, SL projection with a 120° FOV has been developed using vectorial electromagnetic diffraction theory[35]. The intensities of the diffracted beams are optimized using an interior-point method but have a limited number of diffracted beams, up to ±4 diffraction orders. Moreover, 441 random point generating metasurfaces have been demonstrated, underlining the scalable nanofabrication of metasurfaces on 12-inch glass wafer using immersion lithography[36]. However, the largest scanning angle is limited to 15°. The metasurfaces projecting dots of light into diffraction angles close to

90°, giving a 180° FOV, have been demonstrated in the reflective and transmissive spaces[37,38]. The diffraction efficiencies to both reflective and transmissive spaces are same (27%) and were experimentally verified by a blazed grating and beam splitter. As a result, the 4044 projected dots are generated from the pixelated metasurfaces. Furthermore, such metasurface-based SL imaging with a large FOV can be integrated with on-chip light sources such as vertical cavity surface-emitting lasers (VCSELs), facilitated by the flatness features of the metasurfaces[39–41].

Here, we propose a metasurface-based SL projecting device with ~10 K of high-density dots of light or ~100 of parallel lines of light into the full space, realizing an extreme FOV of 180° (Fig. 1). The metasurfaces are composed of periodic supercells designed to diffract incident laser light to high-density number of diffraction orders with uniform intensity. Then, the supercells are periodically arranged with different periodicities along the $x$- and $y$-directions with consideration of the interference effects induced from the arrangement of supercells. The final diffraction patterns are analyzed by the convolution theorem regarding the phase-only distribution of a single supercell as the kernel function of the convolution. By understanding the total optical response with respect to the diffraction and interference effects[42–44], multiple illumination types of the metasurface-based SL projector, i.e., 2D dot arrays, 1D dot arrays, and 2D parallel line arrays of light are numerically and experimentally demonstrated covering a 180° FOV. As a proof-of-concept, we place face masks within the range of 1 m with wide viewing angle up to 60° with respect to optical axis and illuminate the high-density dot arrays generated from the proposed metasurfaces. The depth information of the 3D face masks is extracted from stereo matching algorithm using two cameras. Furthermore, we demonstrate a prototype of the metasurface-based depth sensor for compact and lightweight AR glasses using a nanoparticle-embedded-resin (nano-PER)-based scalable imprinting fabrication method, facilitating direct printing of metasurface to a curved surface of glasses. Such a metasurface-based SL imaging platform can realize imaging of 3D objects over a full FOV with high-density dot arrays in an ergonomically and commercially viable form factor.

## Results

### Design principle of the full-space diffractive metasurface

The diffraction pattern of periodic scatterers can be analyzed by two effects: the diffraction effect of single scatterer and interference effects induced by the periodic arrangement of the scatterers. This viewpoint is widely adopted to understand the optical response of periodic structures, e.g., diffraction from many slits[45], X-ray diffractometry[46], and light-emitting metasurfaces[42]. The interaction between neighboring scatterers is assumed to be negligible to comply

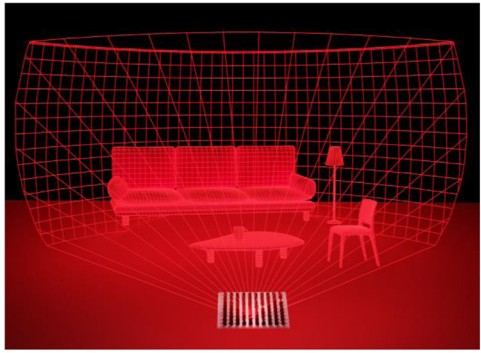
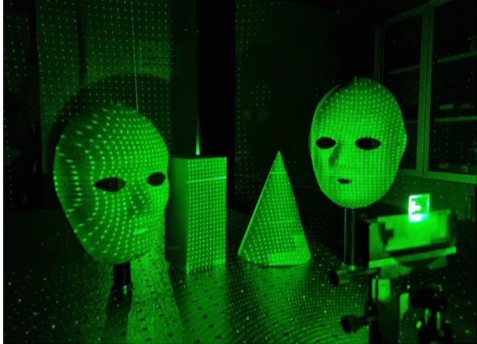

**Fig. 1 | Metasurface-based SL imaging platform scattering high-density diffracted beams into the full 180° FOV.** Under the illumination of a polarization-independent coherent laser source, the proposed metasurfaces generate ~10 K points over the entire 180° space. The depths of the dot arrays illuminated on the objects are extracted using a stereo matching algorithm.

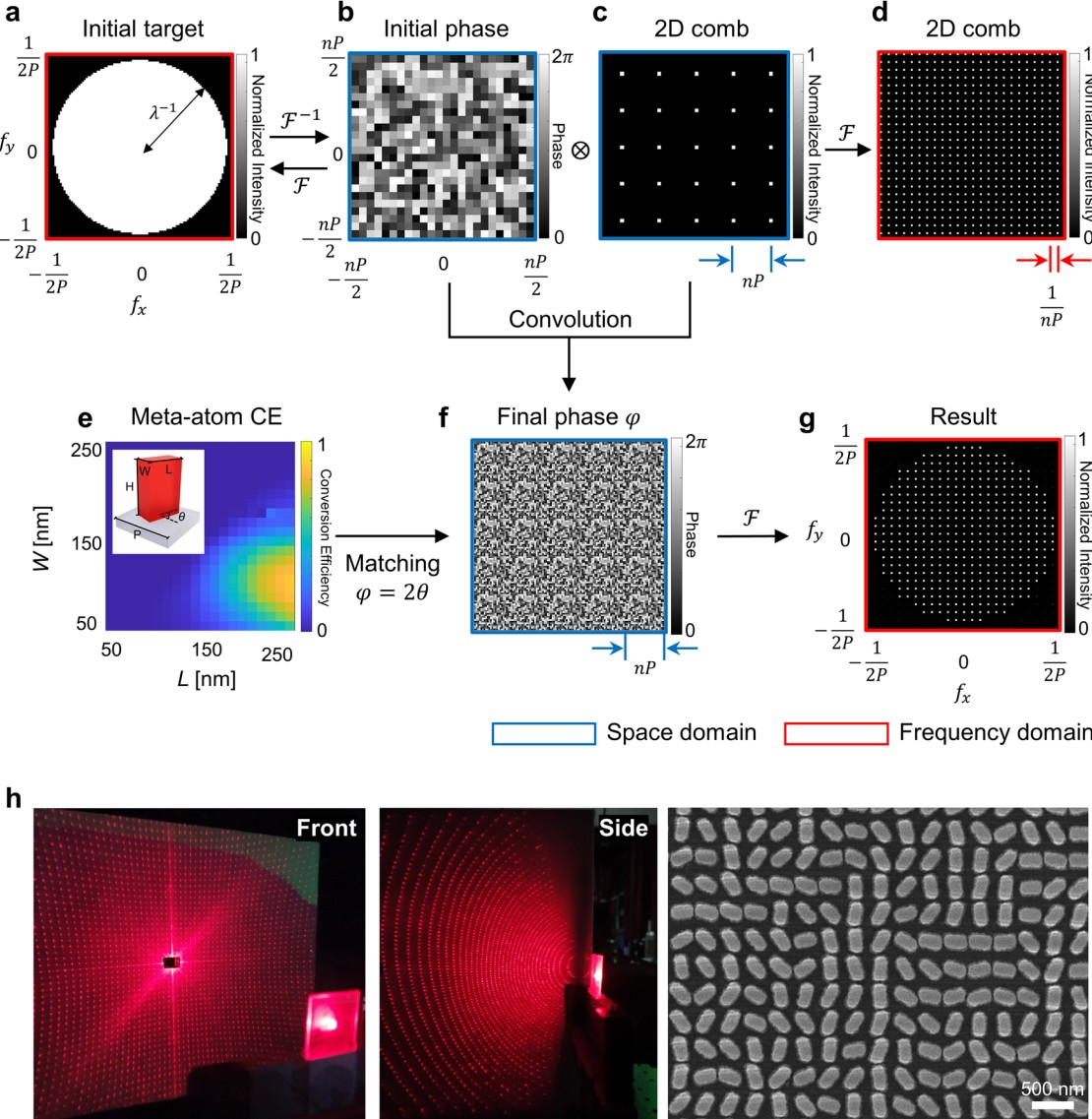

**Fig. 2 | Design principle and experimental demonstration of full-space diffractive metasurface. a–g** Flowchart of designing the metasurfaces. The color of the border, blue and red, represents the space and frequency domain, respectively. **a** Desired spatial frequency distribution. All the propagating waves are designated to have uniform amplitude, excluding excited evanescent waves from the target distribution. **b** The kernel function of the convolution. From the target spatial frequency distribution, the phase-only plate, ranging from 0 to $2\pi$, is retrieved from an iterative Fourier transform algorithm, namely the GS algorithm. **c** 2D Dirac comb function at the space domain representing the supercell arrangement. **d** Fourier transformed of 2D Dirac comb function with period of $1/nP$. **e** The

building blocks of the geometric phase-based metasurfaces. Simulated CE of the meta-atom using RCWA as a function of length ($L$) and width ($W$) with fixed height ($H$) and pitch ($P$). **f** Final phase $\varphi$ obtained from convolution of kernel function and 2D Dirac comb function at the space domain. **g** The final diffracted pattern representing discrete order of diffraction, which is same with multiplication of initial target (**a**) and 2D Dirac comb at the frequency domain (**d**) according to the convolution theorem. **h** Experimental demonstration of the full-space diffracting metasurfaces at the observation plane placed front and side of the metasurface with a scanning electron microscopy (SEM) image of the fabricated metasurfaces.

with Huygens' principle, where each point of the wavefront is considered a secondary source of propagation. The proposed 2D full-space diffractive metasurface is composed of periodically arranged supercells, where each supercell is composed of $n \times n$ meta-atoms with pixel pitch $P$, and $N \times N$ supercells forming the entire metasurface. According to the Nyquist sampling theorem in signal processing, the sampling period should be shorter than half of the signal period to sufficiently resolve the high frequency components. Analogously, the metasurface with pixel pitch of $P$ can resolve spatial frequency $f_{x,y}$ up to $\pm\frac{1}{2P}$, and when the light diffracts at a large angle to the optical axis, it carries high spatial frequency reaching maximum of $\pm\frac{1}{\lambda}$ (Fig. 2a). Therefore, $\pm\frac{1}{\lambda} < \pm\frac{1}{2P}$ gives the upper limit to pixel pitch as $2P < \lambda$. A further effect of the pixel pitch on the diffraction behavior also in

terms of the diffraction grating equation is discussed in Supplementary Note 1. Figure 2a shows the required spatial frequency region, where all propagating waves are located in the radius of $\lambda^{-1}$ have unity amplitude, and evanescent waves located in the $f_x^2 + f_y^2 \geq \lambda^{-2}$ regions are excluded. From the initial target, the phase profile of the single supercell (Fig. 2b) is retrieved using an iterative discrete 2D Fourier transform, i.e., Gerchberg–Saxton (GS) algorithm.

Next, we need to consider interference effects induced by the periodic arrangement of the designed supercells. Figure 2c shows the periodic arrangement itself in both the $x$-, and $y$-directions in the space domain, which is described by a 2D Dirac comb function. Here, the spacing of the comb function is the same as the width of the supercell in the both $x$-, and $y$-directions which is the product of the number of

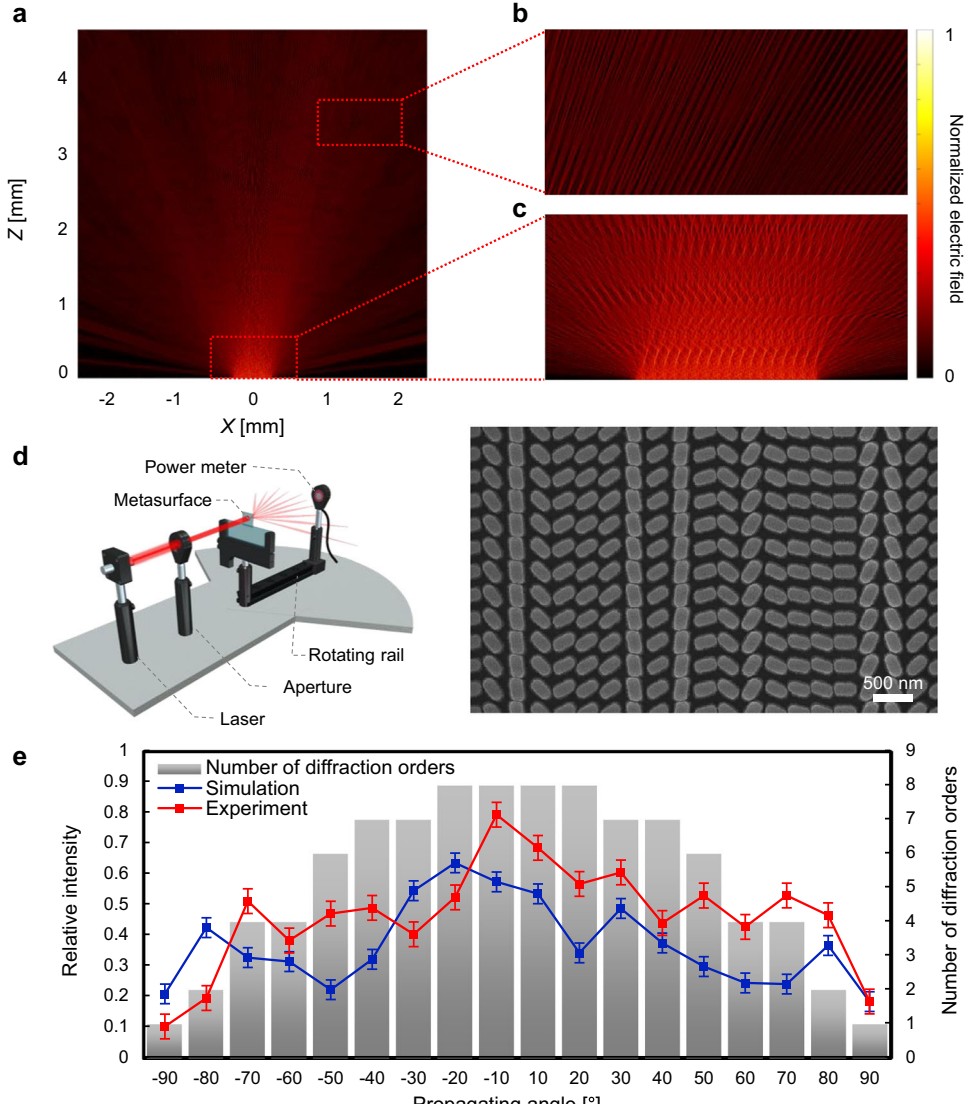

**Fig. 3 | Comparison between simulated and measured diffraction patterns of 1D diffractive metasurfaces. a–c** Simulated distribution of amplitude of electric field at the plane normal of the metasurface (**a**) together with magnifications of the far (**b**), and near fields (**c**). **d** Schematic of the optical setup to measure power of each point by rotating power meter at the operation wavelength of 633 nm where the incident laser light is chirped to fit into the size of metasurfaces. SEM image of the corresponding metasurfaces. **e** Plot of number of diffraction orders, and relative intensities of simulated and measured diffracted beam with respect to propagating angle. The intensities are plotted with averaged value with error bars, as propagating angle is grouped at intervals of 10°.

meta-atoms $n$ and pixel pitch $P$. The entire metasurface is composed of periodically arranged supercells and can be described by the convolution of single supercell and a 2D Dirac comb function resulting in the final phase profile $\varphi$ (Fig. 2f), where the single supercell acts as the kernel function of the convolution. According to the convolution theorem, the Fourier transform of two convolved functions is simply expressed as a multiplication of the Fourier transform of each function. Therefore, the diffraction pattern of the entire metasurface (Fig. 2g) is represented in the spatial frequency domain as a product of the single supercell diffraction pattern (Fig. 2a) and the Fourier transform of 2D Dirac comb function (Fig. 2d). As the Fourier transformed 2D Dirac comb has a spacing of $\frac{1}{nP}$, the maximum signal frequency of $\pm\frac{1}{\lambda}$ is sampled with $\frac{1}{nP}$. Accordingly, the number of diffraction orders is described as $\pm\frac{nP}{\lambda}$, which is $\pm 47$ for $n = 100$, $P = 300$ nm, and $\lambda = 633$ nm. Using a larger supercell with a larger n, the number of diffraction orders can be increased. It should be noted that the intensity of the diffracted beam decreases at large propagating angles because the number of sampling points decreases at higher frequency components. The decrease in intensity of higher-order

diffraction becomes larger as pixel pitch increases, in other words decreasing the uniformity of intensity distribution (Supplementary Fig. 1). The propagation angle of the $m^{th}$ order diffracted beam $\theta_z^m$ with respect to the optical axis is described as $\sin^2\theta_z^m = \sin^2\theta_x^m + \sin^2\theta_y^m$, where angles about the $x$-, and $y$-axis are expressed as

$$\theta_{x,y}^m = \arcsin\left(\frac{m\lambda}{nP}\right), \quad (1)$$

where $m$ represents the diffraction order and the largest diffraction angle $\theta_{x,y}^{m=\pm 47}$ is 82.6°. It should be noted that the uniformly sampled spatial frequency, as shown in Fig. 2g, does not ensure a uniformly spaced propagation angle, since $\sin\theta$ can be approximated to $\theta$ only at small angles. Therefore, the spacing between diffracted beams at higher orders becomes larger, as does the size of the beam. The angle cone of each diffracted beam $\Omega_{N\,x,y}^m$ is described as $\Omega_{N\,x,y}^m = \arcsin\left(\frac{m\lambda}{nP} + \frac{\lambda}{2NnP}\right) - \arcsin\left(\frac{m\lambda}{nP} - \frac{\lambda}{2NnP}\right)$, where $N$ is number of repeated supercells (Supplementary Fig. 2). In addition, the propagation angle is dependent on the operation wavelength; in other words,

the diffraction angle becomes larger at the longer wavelength (Supplementary Fig. 3).

To implement the final phase $\varphi$, the meta-atoms that make up the metasurface should be optimized to modulate the phase of the transmitted light with uniform and high transmission efficiency. For this purpose, we use a rectangular meta-atom made of hydrogenated amorphous silicon (a-Si:H) with high refractive index ($n = 2.8$) and low extinction coefficient ($k = 0.001$) at the wavelength of 633 nm (Supplementary Fig. 4). Under circularly polarized (CP) light incidence, which is represented as $\mathbf{E}_{in} = \begin{bmatrix} 1 \\ \pm i \end{bmatrix}$, the output electric field $\mathbf{E}_{out}$, transmitted through the meta-atom exhibiting strong birefringence along the long and short axes can be described as

$$\mathbf{E}_{out} = \frac{t_l + t_s}{2} \begin{bmatrix} 1 \\ \mp i \end{bmatrix} + \frac{t_l - t_s}{2} e^{\pm i2\theta(x,y)} \begin{bmatrix} 1 \\ \pm i \end{bmatrix}, \qquad (2)$$

where $t_l$ and $t_s$ denote the complex transmission coefficients along the long and short axis of the meta-atom, respectively, and $\theta(x,y)$ is in-plane rotation angle of the meta-atom, as shown in inset of Fig. 2e. From the second term of Eq. (2), the number of oscillations along two axes of nanopillar differs by half of wavelength (Supplementary Fig. 5), manifested by oppositely converted handedness of CP light. According to the physical mechanism of geometric phase, the phase of the converted CP light is modulated by twice of in-plane rotation angle, where + and − sign represents the right circularly polarized (RCP) light, and left circularly polarized (LCP) light incidence, respectively. Therefore, a high polarization conversion efficiency (CE), $|\frac{t_l - t_s}{2}|^2$, is required and is optimized by rigorous coupled-wave analysis (RCWA) as a function of structure parameters, i.e., height, length, and width of the rectangular meta-atom as shown in Fig. 2e. A high CE of 88% is obtained from a meta-atom with $L = 250$ nm, and $W = 110$ nm with $H = 475$ nm. It should be noted that in general, the opposite sign of modulated phase depending on handedness of CP light leads to the unwanted conjugated orders of diffraction. However, in this work, the diffracted beams are designed to propagate in every diffraction order symmetrically to each other (Fig. 2g), in that, it is operable under both RCP and LCP illuminance. As a result, the proposed full-space diffractive metasurface is independent of the incident polarization state, simplifying the optical setup without additional optical components such as polarizer and waveplate.

## Characterization of full-space diffractive metasurface

The final phase distribution obtained in Fig. 2f is realized using the optimized meta-atom, and corresponding subwavelength modulation of light properties enables an extremely large FOV about 180° (Fig. 2h). We also demonstrate a large FOV of our full-space diffractive metasurfaces with a hemisphere screen and Fourier microscope which allows imaging of the spatial frequency domain (Supplementary Fig. 6). We measure the overall diffraction efficiency (DE) to be 60%, defined as all the transmitted light, including zeroth-order beam, normalized to the intensity of the incident laser. The zeroth-order efficiency itself is 32% of the incident light. Compared to the simulated results, we attribute discrepancies to the fact that the fabricated metasurface could not fully realize the calculated phase profile due to the coupling between meta-atoms and fabrication defects. Coupling between neighboring meta-atoms can be alleviated by strongly confining light in the meta-atoms or considering the coupling during the design process of the phase profile. For example, high refractive index titanium dioxide (TiO₂) material-based metasurfaces can strongly confine the electromagnetic waves, reducing the unwanted zeroth-order beam[47]. On the other hand, vectorial diffraction theories such as finite-difference time-domain (FDTD) and Fourier modal method (FMM) can be directly used to calculate the phase profile, which considers the coupling between unit cell in design process[35]. The tilted sidewall profile of fabricated meta-atoms reduces the DE compared to

the calculated CE of 88%. To correct a sidewall profile as right-angled, the etching processes should be carefully optimized to fully exploit the calculated CE[48].

We numerically and experimentally analyze the intensity uniformity of diffracted beams using 1D full-space diffractive metasurface as there are too many points for 2D full-space diffractive metasurface. We simulate a 1D metasurface with the designed structural parameters and material properties using COMSOL Multiphysics, and the electric field distribution in the near field of metasurface is extracted and used as the source of far-field diffraction up to 5 mm from the metasurfaces. Note that this electric field distribution is the full-wave result considering all light-matter interactions at the nanometer scale. After light transmits through the metasurface, it propagates through free space. Therefore, we calculate the amplitude of the electric field distribution in the $x$–$z$ plane using diffraction theory. As the diffracted beams from our metasurface propagate at large angles with respect to the optical axis, the Fresnel or Fraunhofer diffraction theory which uses the paraxial approximation is not appropriate. Therefore, we use Rayleigh–Sommerfeld diffraction theory to visualize wide-angle diffracted light pattern, computed by the fast Fourier transform-based direct integral method[49]. As a result, the amplitude of the electric field is achieved in the $x$–$z$ plane (Fig. 3a). The enlarged far field amplitude distribution shows the high-density diffracted beam paths (Fig. 3b) and the enlarged near field result shows the interference effects between the neighboring supercells (Fig. 3c). Finally, the intensity distribution is obtained over every diffraction order, note that for each diffracted beam, the intensity is numerically integrated considering its finite angular width.

For experimental measurement of intensity of diffracted beams, we fabricate another sample which generates full-horizontal-space dot patterns. To diffract the incident beam into the horizontal direction only, supercells are needed to be designed with different widths along the $x$- and $y$-directions, and convoluted with different periodicities in each direction, equal to the widths of the supercell in $x$- and $y$- directions, respectively (Supplementary Fig. 7a). The power meter is mounted on the rotating rail measuring every diffraction order spanning from −90° to +90° (Fig. 3d). The number of diffraction orders and the intensities of the measured diffracted beams are plotted with the simulated intensity distributions as shown in Fig. 3e. The total ± 47 diffraction orders are grouped into propagating angles at intervals of 10°, and the intensities of the simulated and measured diffracted beams at each group are averaged. As analyzed from Eq. (1), due to the larger angular spacing at high diffraction orders, the number of diffracted beams decreases. The uniformity of the intensity distribution is calculated by comparing the simulation results and the measured values. We obtain an RMSE of 27.48% from the 1D full-space diffractive metasurface, where $RMSE = \sqrt{\sum_{i=1}^{M} \frac{(I_i^{Sim} - I_i^{Exp})^2}{M}}$. Here, $I_i^{Sim}$ and $I_i^{Exp}$ are the simulated and measured $i$th order diffraction intensity, respectively, and $M$ is the total number of the diffraction order. Furthermore, we compare the performance metrics of our full-space diffractive metasurface with state-of-the-art commercial DOE products (Supplementary Note 2).

## Performance of the proposed metasurface-based full-space SL imaging

To demonstrate the prospective application of our metasurfaces as a depth estimating module in 3D sensing technology, we showcase an experimental example of 3D depth reconstruction of a face mask using full-space dot patterns. We use a stereo system with two cameras to calculate the depth of the projected dot points (Fig. 4a). Before the actual measurement, the cameras are calibrated using a standard checkerboard pattern[50] (Supplementary Fig. 8). The intrinsic parameters, e.g., lens focal length, and the extrinsic parameters, e.g., the

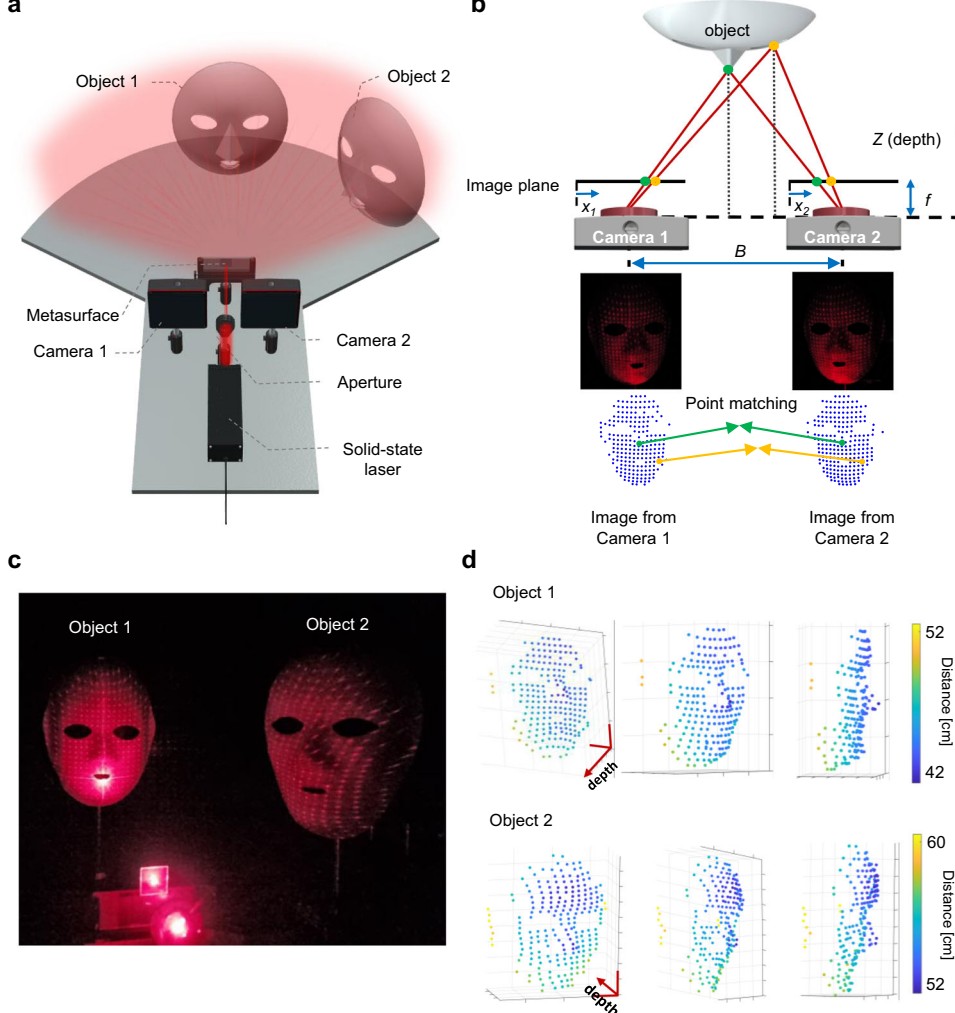

**Fig. 4 | Depth estimation of 3D objects. a** Schematic of the optical setup for depth estimation using 2D dot arrays scattered on objects. **b** Schematic of the stereo matching algorithm used in (**a**). The algorithm uses camera trigonometry and the coordinate difference of the dot points in two image planes. **c** A photograph of the objects; one placed normally and the other placed at 50° with respect to laser beam axis. **d** 3D depth reconstruction results of objects in (**c**) in different rotation view. The first and second row are the depth images of object 1 and object 2, respectively. The depths are presented in colors with respect to the image plane of camera described as the red arrows.

relative location and orientation of each camera, are obtained by detecting the image projection of the pattern. These parameters allow us to map real-world distances to pixels which, in principle, is determined by the relation of similar triangles, $\frac{x_1 - x_2}{f} = \frac{B}{Z}$ in a linear camera model (Fig. 4b). Here, $Z$ is the depth and $x_1$, $x_2$ are the coordinate points in each image plane which correspond to the same point in the object scene. $f$ is the focal length of the camera and $B$ is the distance between the two cameras. The objective is to estimate $Z$, which is inversely proportional to the disparity, $x_1 - x_2$. In the optical setup shown in Fig. 4a, as laser light passes through the metasurface, it projects a set of dots over a wide FOV. The scattered dot points on the surface of the object are captured by each camera.

The depth estimation takes three steps. (1) 2D coordinate extraction of the dots from the two captured images. (2) Matching each dot in camera 1 and its corresponding dot in camera 2 (Fig. 4b). (3) Calculating the depth using the stereo camera trigonometry with the coordinate information of the matched dot pairs. In step 1, the captured images are Gaussian-blurred and binarized and then the resultant contours are detected to determine the coordinate of each dot. In step 2, to match each corresponding dot pair, we use a point set registration method called coherent point drift (CPD) (Supplementary Note 3)[51]. Due to its probabilistic modeling approach, CPD is found to

be robust against outliers and missing points and can preserve the topological structure of the point sets. Note that the CPD algorithm has to compute the inversion of a $M \times M$ matrix per iteration and therefore has a computation complexity of $O(M^3)$ in the case of a non-rigid transformation, where $M$ represents the number of points. In step 3, based on the matched point pairs and the obtained camera parameters, the depth of each dot is calculated using the camera trigonometry relation.

Based on the stereo matching algorithm and by taking advantage of full-space diffractive metasurfaces, a wide FOV 3D sensing is demonstrated. Here, we use a 6 mW-power laser source. Due to the limited power of the laser and the power dispersion of the high-density diffraction orders, the beam intensity drops as the distance and the viewing angle from the metasurface get larger. In addition, because the diffracted beam spreads out in full hemisphere, the spacing between the dots expands as it gets further away from the metasurface, degrading the resolution for faraway objects. Here, the depolarization effect at large-angle diffraction is also considered which distorts the beam shape into elliptical shape (Supplementary Note 4). The vectorial Debye theory can be considered to alleviate distortion, which originates from the deviation of the phase distribution between the metasurface plane and spherical wavefront[10-12]. Therefore, we characterize

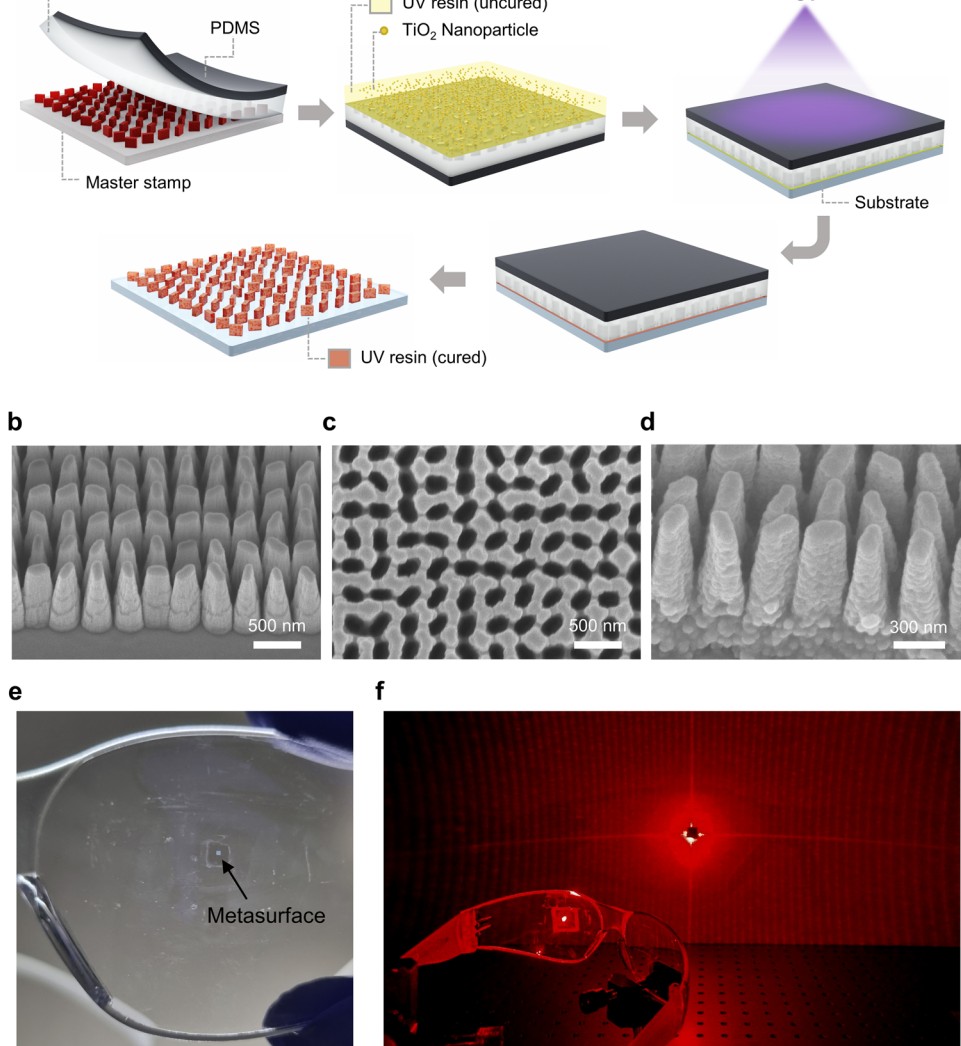

**Fig. 5 | Demonstration of prototype device by direct printing of nano-PER on curved surface of glasses. a** The flow chart of nano-PER based imprinting fabrication process. **b**–**d** SEM image of the master stamp fabricated by EBL (tilted view) (**b**), soft mold composed of *h*-PDMS and PDMS (**c**), and replicated nano-PER-based metasurface (tilted view) (**d**). **e** Replicated metasurface on the curved surface of safety glasses. The size of printed metasurface is 510 μm by 510 μm. **f** Experimental demonstration of full-space covering SL of 2D line array. Laser light illuminate the nano-PER-based metasurface printed on surface of glasses exhibiting application for ultra-compact depth sensor for AR glasses.

the system with maximum axial working distance as 1 m and maximum viewing angle as 60° from optical axis. Although the system is not suitable for objects placed in ranges above these values, such limitations can be overcome using a higher-power laser and designing the metasurface with larger periodicity of the supercells.

We reconstruct the depth of two objects; one placed normally with respect to the laser beam axis, and the other placed at 50° from the axis (Fig. 4c), which shows the wide FOV characteristic of our system while maintaining its depth reconstruction quality. In both cases, the reconstruction of the 3D points in Fig. 4d shows a well-preserved shape of the original face mask with absolute depth error of 8.5% and 1.9% for the object 1 and 2, respectively. In addition, from the reconstruction results, we calculate the relative difference between the maximum and minimum depth point which is 8.4 cm for object 1 and 6.5 cm for object 2. Considering the fact that the actual depth difference (distance from the nearest to the farthest point of face mask) of both objects is 8 cm, the estimation results are fairly accurate. In case of object 2, the relative depth difference is much shorter. This is because of the shaded, sparse projection of dots which is inevitable for objects placed in large FOV, where the dots cannot be projected at shaded, deep surface.

The time taken for the stereo matching algorithm to reconstruct one full-frame depth map was 0.2436 s and 0.3466 s for object 1 and object 2 respectively. This corresponds to roughly 3 to 4 fps processing speed, which is fairly slow compared to the minimum of 30 fps in real-time operating devices. Because of the inherent computational complexity of the CPD algorithm, the calculation time is largely dependent on the number of points. Therefore, the speed of the depth reconstruction process will be slower if there are a greater number of points. Such limitations can be alleviated by adopting other faster algorithms[52]. We did not perform any continuous depth reconstruction task for moving objects in this work due to a lack of the required experimental setup. However, in principle, as long as the number of processing point pairs is comparable to 300 (which was the maximum number of captured points in the experiments), it is possible to operate our system in the scale of 1 fps speed with support from high-speed steaming cameras and an integrated software framework.

## Prototype demonstration with scalable nanofabrication

We demonstrate a prototype of the metasurface-based SL imaging device using a scalable nano-PER imprinting fabrication. Unlike conventional electron beam lithography (EBL) based metasurface fabrication, nanoimprinting can be easily used for high-throughput replication of metasurfaces on any arbitrary (e.g., flat, curved, flexible) substrates. Here, we design a metasurface to generate parallel line patterns for depth extraction. These line patterns are widely used in SL 3D scanning technology[9]. When a single strip of line is projected onto the object scene, the depth information of the object on that line can be reconstructed by intersecting the projection line of the examined image point[53]. SL enables 3D reconstruction without line-by-line scanning by projecting a known parallel set of line pattern onto the object scene and measuring how much it has been altered by the object. The correspondence matching between the elements in the projected and captured patterns requires an encoding of the pattern, and hence, numerous encoding algorithms using time-multiplexing[54–57], line-shifting[58], and dynamic scenes[56,59] have been studied. The proposed full-space diffractive metasurface can be designed to project patterns needed in these algorithms. By applying the convolution theorem with differently designed kernel functions, i.e., the phase profile of the supercell, an arbitrary period and widths of the projected lines can be achieved by arranging the supercells with different periodicities as demonstrated in Supplementary Fig. 7b.

After designing the metasurface, we used soft mold with nano-PER to transfer the pattern of the metasurface to a curved surface of a pair of safety glasses. As shown in Fig. 5a, the master stamp is fabricated using EBL (Fig. 5b) and is then coated with a soft mold composed of hard polydimethylsiloxane (h-PDMS) followed by polydimethylsiloxane (PDMS). The ultraviolet (UV) curable nano-PER is then spin-coated on the soft mold (Fig. 5c). Here, $TiO_2$ nanoparticles are used due to their low extinction coefficient at 633 nm (Supplementary Fig. 4). However, due to the lower refractive index of $TiO_2$ nano-PER compared to a-Si:H at the operation wavelength, the height of nano-PER-based metasurface is designed as 900 nm as shown in tilted view of master mold (Fig. 5b), which is higher than 475 nm height of a-Si:H based metasurface. After putting the nano-PER coated soft mold on the curved surface of glasses, the nano-PER is cured under an UV light source. Finally, the replication of the metasurface is completed on the curved surface after releasing the cured resin (Fig. 5d). Note that the release process of the mold is enhanced by increasing surface tension of the target curved surface using oxygen ($O_2$) plasma and decreasing the surface tension of the soft mold using fluorosurfactant coating.

By illuminating laser light to the nano-PER-based metasurface printed on glasses (Fig. 5e), 2D line array pattern is experimentally demonstrated to cover the full-space as shown in Fig. 5f. Here, the size of printed metasurface is 510 μm by 510 μm. Note that, due to the nonlinearity between spatial frequency and propagation angle described as Eq. (1), line pattern on the flat observation plane shows curved shape at large angle with respect to optical axis. Although we do not provide an explicit demonstration of depth estimation using these patterns, we highlight the versatility of our proposed metasurface with scalable nanofabrication methods in that it can be developed to generate the different patterns required for existing 3D reconstruction algorithms.

## Discussion

In conclusion, we numerically simulated and analyzed the optical response of a metasurface for 3D depth reconstruction, and experimentally validated the full-space diffracted features of the proposed metasurface. Regarding the supercell as the kernel function of the convolution, the diffraction pattern of the entire metasurfaces composed of periodic supercells was analyzed using the convolution theorem, realizing the number of desired diffracted SL patterns needed for 3D object recognition. As a result, a flat metasurface-based SL imaging system has been realized with high-density order of diffraction covering a 180° FOV. In a single-shot of projected dots, the depth information of 3D objects placed at both normal and wide angles was extracted by matching the monitored points at two different positions using a stereo matching algorithm. The demonstrated stereo system can also adopt other unique, optimized patterns with designed metasurfaces, for further enhancement of their robustness and accuracy[60]. Moreover, we demonstrate a prototype of AR glasses by printing SL projecting metasurface on a curved surface of glasses using the nano-PER imprinting method which enables high-throughput manufacturing of the metasurfaces on any arbitrary substrates.

In terms of operating distance, the proposed metasurface-based SL imaging has limited range compared to single laser scanning, due to the power dispersion through the spreading of light to the multiple diffraction orders. In addition, the spacing between diffracted beams is increased at farther distance from metasurface and at larger angle from optical axis, resulting in limited resolution of imaging due to sparse dot patterns on the object. In this work, the incident power to the metasurface was 6 mW at the operation wavelength of 633 nm, generating ~10 K dots. We achieved operating range of 1 m from metasurface, and 60° from optical axis, i.e., 120° FOV, even though the diffracted SL patterns covered 180° FOV. The operating range can be enhanced using high power incident laser or higher density of dot patterns, which can be achieved by increasing the number of meta-atoms comprising supercell. We experimentally demonstrate the operation of metasurface-based SL imaging reaching long distances with a high power (100 mW) laser (Supplementary Fig. 10). However, considering the safety issues, the invisible near-infrared (NIR) light source at 1550 nm can be considered, where this range of light is strongly absorbed at the cornea and lens, therefore do not reach to the retina. To do so, silicon nanoparticles[61], which have a high refractive index and low extinction coefficient at the NIR can be used instead of the $TiO_2$ nanoparticles, enabling high power operation at longer distances. Furthermore, such a flat passive metasurface-based SL imaging does not require the application of a line-by-line voltage bias compared to the TCO[6] and LC[7,8] based active beam steering approaches, alleviating the requirement of extremely complex wiring configurations. Also, by taking advantage of the flat feature of the proposed metasurfaces, light engines such as VCSELs can be integrated, realizing on-chip illuminating devices for ultra-compact and lightweight 3D imaging systems.

## Methods

### Metasurface fabrication

Metasurface was fabricated on a glass substrate. A 475 nm thick a-Si:H layer was deposited using plasma enhanced chemical vapor deposition (PECVD, BMR Technology HiDep-SC) with silane ($SiH_4$) and hydrogen ($H_2$) gases at a flow rate of 10 sccm and 75 sccm, respectively, at the optimized 25 mTorr pressure and 200 °C. The designed pattern of the metasurface was transferred to positive tone photoresist of polymethyl methacrylate (950 PMMA A2, MicroChem) using a standard EBL process (ELIONIX, ELS-7800; acceleration voltage: 80 kV, beam current: 100 pA). After the development process using MIBK/IPA 1:3 developer, 40 nm thickness chromium (Cr) was deposited using electron beam evaporator (KVT, KVE-ENS4004) and lift-offed. Using Cr patterns as an etching mask, the metasurface pattern was transferred to the a-Si:H layer by dry etching process (DMS, silicon/metal hybrid etcher). Finally, the remaining Cr mask was removed using Cr etchant (CR-7).

### Synthesis of $TiO_2$ nanoparticle embedded resin

The $TiO_2$ nanoparticles were dispersed into MIBK (DT-TIOA-30MIBK (N30), Ditto technology), monomer (dipentaerythritol penta-/hexa-acrylate, Sigma-Aldrich), photo-initiator (1-Hydroxycyclohexyl phenyl ketone, Sigma-Aldrich), and MIBK solvent (MIBK, Duksan general

science). The weight ratio was selected as 4, 0.7, and 0.3 wt% for $TiO_2$ nanoparticles, monomer, and photo-initiator, respectively.

## Measurements

To characterize the scattering properties of the full-space diffractive metasurface, we demonstrated an optical setup as shown in Fig. 4a. The incident light from the solid-state He–Ne laser (HNL210L, Thorlabs), with wavelength of 633 nm and power of 6 mW, was cropped by pinhole aperture to fit into the metasurface. When the light passes through the metasurface, the high-density dot patterns are generated and projected onto the full-space. The scattered points from the surface of the object were captured by two cameras, which were positioned in parallel with a slight coordinate difference.

## Depth information extraction

Stereo cameras were calibrated using MATLAB camera calibrator application. Images of the standard checkerboard pattern in 15–20 different angle and positions were taken by two cameras respectively. Then, effective camera parameters are retrieved by detecting and matching the points on the pattern. High-error images were excluded to improve calibration. The depth of each dot was calculated using the Image Processing and Computer Vision APIs in MATLAB.

## Data availability

The data that support the findings of the study are available from the corresponding author upon reasonable request.

## Code availability

The code used for the metasurface design is available from the corresponding author upon reasonable request.

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

## Acknowledgements

This work was financially supported by the Samsung Research Funding & Incubation Center for Future Technology grant (SRFC-IT1901-52) funded by Samsung Electronics, the POSCO-POSTECH-RIST Convergence Research Center program funded by POSCO, and the National Research Foundation (NRF) grant (NRF-2022M3C1A3081312) funded by the Ministry of Science and ICT (MSIT) of the Korean government. G.K. acknowledges the POSTECH Alchemist fellowship. Y.K. acknowledges the Hyundai Motor *Chung Mong-Koo* fellowship, and the NRF Ph.D. fellowship (NRF-2022R1A6A3A13066251) funded by the Ministry of Education of the Korean government. Y.K. and J.Y. acknowledge the NRF International Research & Development fellowships (NRF-2022K1A3A1A12080445 and NRF-2022K1A3A1A12080092), respectively, funded by the MSIT of the Korean government. I.K. acknowledges the NRF *Sejong* Science fellowship (NRF-2021R1C1C2004291) funded by the MSIT of the Korean government. The authors thank Dr. Junghyun Park and Dr. Seunghoon Han (Samsung Advanced Institute of Technology) for discussions.

## Author contributions

J.R., I.K., and G.K. conceived the initial idea, and developed the concept. G.K. calculated and designed the full-space diffractive metasurfaces. G.K. and S.K. performed the simulation of diffractive characteristics of proposed metasurfaces. I.K. and G.K. fabricated the metasurface. G.K., Y.K., I.K., and J.P. conducted the optical measurements. J.Y. developed the stereo matching algorithms and performed the depth estimation of 3D objects. Y.K. and J.P. prepared the data sets of 3D objects for depth estimation. J.K. synthesized the $TiO_2$ nanoparticles and conducted replication of metasurface. S.-W.M., S.K., and T.B. provided theoretical advice. All authors wrote the manuscript, contributed to the discussion and analysis. J.R. guided the entire project.

## Competing interests

The authors declare no competing interests.
