## [Peer Review File · Nature Communications]

Metasurface-driven full-space structured light for three-dimensional imagingREVIEWER COMMENTS

Reviewer #1 (Remarks to the Author):

3D imaging/depth imaging is a topic of broad interest, with potential applications in many fields. The paper written by Kim et al. reports metasurface-driven structured light projection and 3D imaging based on the structured light technique. It sets a record on the field of view (FOV) of a diffractive optical element (DOE) for high-density & uniform structured light projection. In addition, I appreciate that the authors went the extra mile to actually demonstrate depth-sensing over a large FOV using the metasurface DOE. At the same time, most previous works stop after characterizing the metasurface itself. Furthermore, they show that the metasurface DOE can be fabricated using a high-throughput printing process. The theory and experiments are both well explained. The paper is timely, should be of great interest to the broader photonics community. Therefore, I recommend this manuscript for publication in Nature Communications, with some suggestions listed below for revision:

1. Since there are already many commercial DOE products for structured light projection (for example: <https://www.holoor.co.il/structured-light-doe/>), I suggest the authors provide quantitative value on the key performance metrics of their metasurface DOE, such as the FOV, overall diffraction efficiency, zeroth-order efficiency, and spot uniformity. Such that the readers may have a better sense of the relative advantages and disadvantages of the metasurface DOE.
2. One issue I see from the experimental results (for example, in Fig. 2h and Fig. 4c) is that the zeroth-order diffraction of the metasurface DOE is quite strong, despite that the calculated zeroth-order diffraction is relatively weak (Fig. 2g). I believe the discrepancy may originate from the fact that the coupling between the unit cell is not considered here, such that the diffraction phase of the fabricated structure may not fully represent what is shown in the designed phase (Fig. 2f). I suggest the authors provide some discussion on how to alleviate the issue.
3. I suggest the authors comment on the time it takes the point-matching algorithm to recover one full frame of the depth map.

Reviewer #2 (Remarks to the Author):

In this manuscript, the authors reported a metasurface-based structured light imaging system, which covers the full 180 degrees field of view and a high-density ~10K dot array. The depth information from backscattered light of some objects using a stereo matching algorithm was experimentally extracted out based on the reported optical system. This demonstration offers a useful prototype of 3D imaging system, which can extract depth information from backscattered light of 3D objects using a stereo matching algorithm. I find these results exciting and believe they may represent an important breakthrough in the use of ultrathin and lightweight metasurface for an advanced 3D imaging system.

Overall, both theoretical calculation and experimental demonstration are sound, the results are of high novelty and may find broad impact on photonics and information optics. I would like to recommend this work for publication in Nature Communications after the authors have addressed my suggestions below.

1. My major comment is on the design of the large supercell array metasurface. I understand that the current design flow in Figure 2, but can the authors compare and articulate why their approach is the best?
2. The metasurface was designed based on the 2D repetition of a supercell metasurface, which is in the form of a phase-only metasurface hologram. According to the authors: "the properties of diffraction patterns are analyzed by convolution theorem considering the supercell as a kernel function", this means the supercell with a size of only 100 by 100 meta-atoms and a pitch of 300 nm was optimised to create a flat-top intensity distribution over the whole imaging plane (Fig. 2a). However, I cannot find the specific results for the characterisation of this small piece metasurface, how about the intensity uniformity, diffraction angle coverage and efficiency?
3. Is the pitch of 30 μm of the supercell metasurface an optimised value? How do the numbers of the

supercell pitch and the number of pixels in the metasurface supercell affect the diffraction of a flat-top target intensity and the number of diffraction spots in the imaging plane? How does the depolarisation effect affect the large-angle vectorial diffraction? Can the authors provide some discussion?

4. Based on the Fourier theorem, multiply the Fourier transform of the supercell (equivalent to a flat top across the whole imaging plane) by a 2D comb function results in an array of multi-order diffracted spots in the imaging plane (Fig. 2g). My major concern is, however, about the efficiency of designing such a small supercell metasurface as the convolution kernel in the metasurface plane. My understanding is that this small supercell needs to consider a large area of diffraction across the whole imaging plane (a few millimeters according to Fig. 3a, or even meters in the experiment), which includes large-angle diffractions. Can the authors discuss why this design approach is the best?

5. I would like the authors at least compare this design approach with another intuitive case, for instance, a small metasurface supercell designed for creating a multifocal array that covers only a small projection area in the imaging plane? In this case, each metasurface supercell can be flexibly designed in terms of spot number and diffraction angle. Compared to the authors' flat top target, how about directly creating a multifocal array with a high uniformity? Like the results reported in the following papers.

[1] Opt. Lett. 36, 406-408 (2011); [2] Opt. Lett. 39, 1621-1624 (2014).

Reviewer #3 (Remarks to the Author):

Recommendation: Accept after revision

This manuscript entitled "Metasurface-driven full-space structured light three-dimensional imaging" reports a metasurface that can generate dot array covering large field of illumination (FOI) of near 180 degrees. The dot array is center-symmetric such that the metasurface consisting of nanofins is dependent to the polarization of incident light. The authors showcase that such metasurface-generated dot array is able to performance depth sensing with a few millimeters error for a face object placed at 1 meter away from the metasurface. Lastly, a nanoimprint was shown as proof of concept for large-scale manufacturing.

I highly recommend this manuscript for publication after the following itemized questions are addressed in the revision. This work is the first demonstration of metasurface-generated dot array with large FOI for depth sensing.

1. What is the efficiency of the metasurface shown in Fig. 2? It seems all metasurfaces have strong 0th order, what is the reason?
2. For the experiment results shown in Fig. 4, was the same metasurface of Fig. 2 used? I am also curious to know the spot size diameter and spot-to-spot distance. Is the metasurface's size larger than the incident laser beam diameter? What is the laser beam divergence? I ask these questions because the diameter and spot-to-spot distance of dot array are vital parameters, which determine lateral spatial resolution. Typically, for face recognition used in mobile phones, the dot diameter needs to achieve ~ 1 mm while projecting the dot array to a screen 20 cm away from the projector.
3. It was mentioned through this manuscript that the metasurface has a field of illumination of 180 degrees; however, I do not see a convincing measurement data to support it. An image taken by a Fourier

microscope (J.Opt.Soc.Am. A 32, 20822092 (2015)) or by projecting the dot array to a hemispherical screen (Adv. Mater. 2012, 24, OP331–OP336) could be a feasible approach.

4. Line 162 on page 6, it should read "... since $\sin(\theta)$ can be approximated to θ only at ...".

Reviewer #1 (Remarks to the Author):

3D imaging/depth imaging is a topic of broad interest, with potential applications in many fields. The paper written by Kim et al. reports metasurface-driven structured light projection and 3D imaging based on the structured light technique. It sets a record on the field of view (FOV) of a diffractive optical element (DOE) for high-density & uniform structured light projection. In addition, I appreciate that the authors went the extra mile to actually demonstrate depth-sensing over a large FOV using the metasurface DOE. At the same time, most previous works stop after characterizing the metasurface itself. Furthermore, they show that the metasurface DOE can be fabricated using a high-throughput printing process. The theory and experiments are both well explained. The paper is timely, should be of great interest to the broader photonics community. Therefore, I recommend this manuscript for publication in Nature Communications, with some suggestions listed below for revision:

We appreciate the reviewer for acknowledging our effort in the demonstration of metasurface-based depth-sensing over an outstanding field of view and high-throughput imprinting process. The raised comments from the reviewer are carefully answered as below.

Comment 1:

Since there are already many commercial DOE products for structured light projection (for example: <https://www.holoor.co.il/structured-light-doe/>), I suggest the authors provide quantitative value on the key performance metrics of their metasurface DOE, such as the FOV, overall diffraction efficiency, zeroth-order efficiency, and spot uniformity. Such that the readers may have a better sense of the relative advantages and disadvantages of the metasurface DOE.

Our response 1:

We thank the reviewer for the helpful suggestions about key metrics that we need to evaluate. As the reviewer suggested, we have evaluated our metasurface compared to the conventional DOE products and previously reported metasurface-based structured light projectors with respect to the diffraction efficiency (including zeroth-order efficiency of our metasurface), spot uniformity, field of view, and number of spots. All metrics are measured from our 2D full-space diffractive metasurface, and the spot uniformity is measured using a 1D diffractive metasurface as there are too many spots for our 2D sample. We reflected the measured metrics in the manuscript and added a comparison of our work and others in Supplementary Note 2 with Supplementary Table 2 as below:

“The uniformity of the intensity distribution is calculated by comparing the simulation results and the measured values. We obtain a RMSE of 27.48% from the 1D full-space diffractive metasurface, where $RMSE = \sqrt{\sum_{i=1}^M \frac{(I_i^{Sim} - I_i^{Exp})^2}{M}}$. Here, I_i^{Sim} and I_i^{Exp} are the simulated and measured i^{th} order diffraction intensity, respectively, and M is the total number of the diffraction order. Furthermore, we compare the performance metrics of our full-space diffractive metasurface with state-of-the-art commercial DOE products (Supplementary Note 2).”

Supplementary Note 2. Comparison of commercial DOE products and metasurface-based structured light projectors.

The key performance metrics of structured light projection are field of view, diffraction efficiency, zeroth-order efficiency, and spot intensity uniformity. The measured FOV of our 2D full-space diffractive metasurfaces reaches 180° with an overall diffraction efficiency of 60%. The zeroth-order makes up 32% of the total incident intensity. To measure the uniformity of the spot intensities we use a 1D full-space diffractive metasurface as there are too many dots in our 2D full-space diffractive metasurface. The root means square error (RMSE) value approaches 27.48%. The values of the key metrics are compared with previously reported metasurface-based structured light projectors²⁻⁴ and commercial DOE products⁵⁻⁸ (Supplementary Table 2). Both diffraction efficiency and spot uniformity of our metasurface are comparable to state-of-the-art DOE products and previously reported metasurface-based structured light projectors, while our 2D full-space diffractive metasurfaces exhibit a record for the FOV in the transmissive regime. It should be noted that the deflection efficiency of 46.5% is measured normalized to the transmitted light due to low transmission, originating from the bonding process of immersion lithography technology for the values in ref³.

Supplementary Table 2. Previously reported metasurface-based structured light projectors and commercial DOE products.

Ref.	Diffraction efficiency (%)	Spot uniformity RMSE (%)	Field of view (°)	Number of spots
Ours	60 (2D)	27.48 (1D)	180	6921 (2D)

[2]	55.6 (1D) 59.1 (2D)	39.71 (1D)* 38.68 (2D)*	120	9 (1D) 69 (2D)
[3]	46.5 (2D) (deflection efficiency)	Not reported	30	441 (2D)
[4]	Not reported	Not reported	360	4044 (2D)
[5]	82.2	Not reported	4.994	144 (2D)
[6]	67	Not reported	30.77	32761 (2D)
[7]	63	Not reported	60 (horizontal) 66 (vertical)	81 (2D)
[8]	Not reported	Not reported	53.3 (horizontal) 67.6 (vertical)	101050 (2D)

* $RMSE = \frac{1}{1/M} \sqrt{\sum_{i=1}^M \left(I_i - \frac{1}{M}\right)^2} / M$ where I_i is the i^{th} order diffraction intensity normalized to the incident light intensity, and M is the total number of diffraction orders.

Comment 2:

One issue I see from the experimental results (for example, in Fig. 2h and Fig. 4c) is that the zeroth-order diffraction of the metasurface DOE is quite strong, despite that the calculated zeroth-order diffraction is relatively weak (Fig. 2g). I believe the discrepancy may originate from the fact that the coupling between the unit cell is not considered here, such that the diffraction phase of the fabricated structure may not fully represent what is shown in the designed phase (Fig. 2f). I suggest the authors provide some discussion on how to alleviate the issue.

Our response 2:

We thank the reviewer for the constructive comments on the discrepancy between experimental (Fig. 2h and Fig. 4c) and calculated results (Fig. 2g), and where it comes from. As the reviewer indicated, the discrepancy comes from the fact that metasurface unit cells could not fully realize the calculated phase profile due to the coupling between them. Therefore, the discrepancy, especially quite strong zeroth-order diffraction, can be alleviated by minimizing the coupling between unit cells or considering coupling at the design process of the phase profile. We have added the discussion on the reason for the discrepancy, and the approaches to alleviate discrepancy to the manuscript as below:

“The final phase distribution obtained in Fig. 2f is realized using the optimized meta-atom, and corresponding subwavelength modulation of light properties enables an extremely large FOV about 180° (Fig. 2h). We also demonstrate a large FOV of our full-space diffractive metasurfaces with a hemisphere screen and Fourier microscope which allows imaging of the spatial frequency domain (Supplementary Figure 6). We measure the overall diffraction efficiency (DE) to be 60%, defined as all the transmitted light, including zeroth-order beam, normalized to the intensity of the incident laser. The zeroth-order efficiency itself is 32% of the incident light. Compared to the simulated results, we attribute discrepancies to the fact that the fabricated metasurface could not fully realize the calculated phase profile due to the coupling between meta-atoms and fabrication defects. Coupling between neighboring meta-atoms can be alleviated by strongly confining light in the meta-atoms or considering the coupling during the design process of the phase profile. For example, high refractive index titanium dioxide (TiO₂) material-based metasurfaces can strongly confine the electromagnetic waves, reducing the unwanted zeroth-order beam⁴⁶. On the other hand, vectorial diffraction theory such as finite-difference time-domain (FDTD) and Fourier modal method (FMM) can be directly used to calculate the phase profile, which consider the coupling between unit cell in design process³⁴. The tilted sidewall profile of fabricated meta-atoms reduces the DE compared to the calculated CE of 88%. To correct a sidewall profile as right-angled, the etching processes should be carefully optimized to fully exploit the calculated CE⁴⁷.”

Comment 3:

I suggest the authors comment on the time it takes the point-matching algorithm to recover one full frame of the depth map.

Our response 3:

We thank the reviewer for this comment. As the reviewer suggested, we have measured the time taken to reconstruct the depth of one full frame in each experimental case and have added an extended discussion on the required calculation time of our system at the manuscript as below:

“Note that the CPD algorithm has to compute the inversion of a $M \times M$ matrix per iteration and therefore has a computation complexity of $O(M^3)$ in the case of a non-rigid transformation, where M represents the number of points.”

“...The time taken for the stereo matching algorithm to reconstruct one full-frame depth map was 0.2436 sec and 0.3466 sec for object 1 and object 2 respectively. This corresponds to roughly 3 to 4 frames/sec processing speed, which is fairly slow compared to the minimum of 30 frames/sec in real-time operating devices. Because of the inherent computational complexity of the CPD algorithm, the calculation time is largely dependent on the number of points. Therefore, the speed of the depth reconstruction process will be slower if there are a greater number of points. Such limitations can be alleviated by adopting other faster algorithms⁵¹. We did not perform any continuous depth reconstruction task for moving objects in this work due to a lack of the required experimental setup. However, in principle, as long as the number of processing point pairs is comparable to 300 (which was the maximum number of captured points in the experiments), it is possible to operate our system in the scale of 1 frame/sec speed with support from high-speed steaming cameras and an integrated software framework.”

Reviewer #2 (Remarks to the Author):

In this manuscript, the authors reported a metasurface-based structured light imaging system, which covers the full 180 degrees field of view and a high-density ~10K dot array. The depth information from backscattered light of some objects using a stereo matching algorithm was experimentally extracted out based on the reported optical system. This demonstration offers a useful prototype of 3D imaging system, which can extract depth information from backscattered light of 3D objects using a stereo matching algorithm. I find these results exciting and believe they may represent an important breakthrough in the use of ultrathin and lightweight metasurface for an advanced 3D imaging system. Overall, both theoretical calculation and experimental demonstration are sound, the results are of high novelty and may find broad impact on photonics and information optics. I would like to recommend this work for publication in Nature Communications after the authors have addressed my suggestions below.

We thank the reviewer for constructive comments on our metasurface-based structured light imaging system. We are delighted that the reviewer finds our metasurface work as important breakthrough for an ultra-compact and advanced 3D imaging system. The reviewer raised several important comments which we have responded to carefully as below.

Comment 1:

My major comment is on the design of the large supercell array metasurface. I understand that the current design flow in Figure 2, but can the authors compare and articulate why their approach is the best?

Our response 1:

We thank for the reviewer for comments on design of metasurface array that have allowed us to articulate the advantages of our approach. The key parameters of metasurface-based structured light imaging system, which determine the resolution of depth sensing, are field of view, number of spots, spot diameter and spot-to-spot distance. Compared to the other metasurface for structured light projector as shown in Supplementary Table 2, our approach allows maximum field of view in transmissive regime and high-density of spots. Field of view is pre-dictated at design of single supercell by selecting the target propagating angle at the spatial frequency domain, and the number of spots is function of number (n) and pitch (P) of the supercell pixel. Then, the spot diameter and spot-to-spot distance can be designed by

modulating the number of supercells (N) to form the periodically arranged metasurface array. Therefore, in terms of depth sensing, the key parameters can be tuned intuitively using few degrees of freedoms. Simultaneously, the diffraction efficiency and intensity uniformity are comparable to the state-of-the-art DOE products as shown in Supplementary Table 2.

Supplementary Table 2. Previously reported metasurface-based structured light projectors and commercial DOE products.

Ref.	Diffraction efficiency (%)	Spot uniformity RMSE (%)	Field of view (°)	Number of spots
Ours	60 (2D)	27.48 (1D)	180°	6921 (2D)
[2]	55.6 (1D) 59.1 (2D)	39.71 (1D)* 38.68 (2D)*	120°	9 (1D) 69 (2D)
[3]	46.5 (2D) (deflection efficiency)	Not reported	30°	441 (2D)
[4]	Not reported	Not reported	360°	4044 (2D)
[5]	82.2	Not reported	4.994°	144 (2D)
[6]	67	Not reported	30.77°	32761 (2D)
[7]	63	Not reported	60° (horizontal) 66° (vertical)	81 (2D)
[8]	Not reported	Not reported	53.3° (horizontal) 67.6° (vertical)	101050 (2D)

* $RMSE = \frac{1}{1/M} \sqrt{\sum_{i=1}^M \left(I_i - \frac{1}{M}\right)^2} / M$ where I_i is the i^{th} order diffraction intensity normalized to the incident light intensity and M is the total number of diffraction orders.

Comment 2:

The metasurface was designed based on the 2D repetition of a supercell metasurface, which is in the form of a phase-only metasurface hologram. According to the authors: “the properties of diffraction patterns are analyzed by convolution theorem considering the supercell as a kernel function”, this means the supercell with a size of only 100 by 100 meta-atoms and a pitch of 300 nm was optimised to create a flat-top intensity distribution over the whole imaging plane (Fig. 2a). However, I cannot find the specific results for the characterisation of this small piece metasurface, how about the intensity uniformity, diffraction angle coverage and efficiency?

Our response 2:

We thank the reviewer for the comments on the design and characterization of supercell. First of all, the notation of the width of supercell (d) is replaced by multiplication of the number of pixels (n) and pitch of pixel (P) to better exhibit the effect of supercells, as a function of n and P , on the four metrics: (1) Intensity distribution of diffracted beams (2) diffraction angle coverage (3) diffraction efficiency, and (4) number of diffracted beams. We have added discussion on design of single supercell, which pre-dictates the four metrics that appear when repeated periodically to form a metasurface array, in manuscript and Supplementary Note 1 as below:

“The proposed 2D full-space diffractive metasurface is composed of periodically arranged supercells, where each supercell is composed of $n \times n$ meta-atoms with pixel pitch P , and $N \times N$ supercells forming the entire metasurface. According to the Nyquist sampling theorem in signal processing, the sampling period should be shorter than half of the signal period to sufficiently resolve the high frequency components. Analogously, the metasurface with pixel pitch of P can resolve spatial frequency $f_{x,y}$ up to $\pm \frac{1}{2P}$, and when the light diffracts at a large angle to the optical axis, it carries high spatial frequency reaching maximum of $\pm \frac{1}{\lambda}$ (Fig. 2a). Therefore, $\pm \frac{1}{\lambda} < \pm \frac{1}{2P}$ gives the upper limit to pixel pitch as $2P < \lambda$. A further effect of the pixel pitch on the diffraction behavior also in terms of the diffraction grating equation is discussed in Supplementary Note 1. Figure 2a shows the required spatial frequency region, where all propagating waves are located in the radius of λ^{-1} have unity amplitude, and evanescent waves located in the $f_x^2 + f_y^2 \geq \lambda^{-2}$ regions are excluded. From the initial target, the phase profile of the single supercell (Fig. 2b) is retrieved using an iterative discrete 2D Fourier transform, i.e., Gerchberg-Saxton (GS) algorithm.”

“...Therefore, the diffraction pattern of the entire metasurface (Fig. 2g) is represented in the spatial frequency domain as a product of the single supercell diffraction pattern (Fig. 2a) and the Fourier transform of 2D Dirac comb function (Fig. 2d). As the Fourier transformed 2D Dirac comb has a spacing of $\frac{1}{nP}$, the maximum signal frequency of $\pm \frac{1}{\lambda}$ is sampled with $\frac{1}{nP}$. Accordingly, the number of diffraction orders is described as $\pm \frac{nP}{\lambda}$, which is ± 47 for $n = 100$, $P = 300\text{nm}$, and $\lambda = 633\text{ nm}$. Using a larger supercell with a larger n , the number of

diffraction orders can be increased. It should be noted that the intensity of the diffracted beam decreases at large propagating angles because the number of sampling points decreases at higher frequency components. The decrease in intensity of higher-order diffraction becomes larger as pixel pitch increases, in other words decreasing the uniformity of intensity distribution (Supplementary Figure 1).”

Supplementary Note 1. The effect of pixel pitch on diffraction behavior.

P_N and P_G are the pixel pitch determined from Nyquist sampling theorem and grating equation, respectively, give criteria for the choice of pixel pitch of the single supercell. The light transmitted through the metasurface can be considered as a signal with a bandwidth of $2k_0$, where k_0 is the free-space wavenumber. If such a band-limited signal is sampled with a sampling frequency of k_N , the signals are added to the spectrum with an interval of k_N , thus k_N should be larger than $2k_0$ to perfectly reconstruct a signal of $2k_0$ bandwidth. In other words, P_N needs to be smaller than half of the free-space wavelength, resulting in $P_N < 316$ nm. On the other hand, high-order diffraction occurs even in a single supercell when the pixel pitch is larger than P_G , as derived from the transmission grating equation described as

$$n_t \sin \theta_m = n_i \sin \theta_i + m \frac{\lambda_0}{P}, \quad (1)$$

where n_t and n_i represent the refractive index of the transmitted and incident media, θ_m and θ_i denote the angles of the m^{th} order diffraction and incident beams, λ_0 is the vacuum wavelength, and P is the pixel pitch. To prevent high-order diffraction in a single supercell under normal incidence, P_G should be smaller than $\frac{\lambda_0}{n_t}$, resulting in the condition $P_G < 633$ nm.

We simulate the effect of pixel pitch on the intensity distribution of the diffracted beams by varying the pixel pitch from 300 nm to 700 nm with an interval of 100 nm with the optimized meta-atom. The range of pixel pitch spans over P_N and P_G . Below 300 nm, the neighboring nanostructures are electromagnetically coupled, in other words the evanescent electromagnetic waves of nanostructures are not diminished before reaching the neighboring nanostructures¹. The number of pixels of a single supercell is selected to allow comparing the intensity of diffracted beams at the same diffraction order (Supplementary Table 1). The pixel pitch can be divided into three regimes: (1) $P < P_N$, (2) $P_N < P < P_G$, and (3) $P_G < P$. The intensity,

angle, and number of diffracted beams which are pre-determined from the single supercell appear when repeated periodically to form a 4 by 4 supercell array (Supplementary Figure 1). The number of supercell arrays is selected as four, due to computational limitations to simulate using the FDTD method. In the first regime, all diffracted beams propagate with a moderately uniform intensity, however, the intensity drops at large angles, which is unavoidable due to the decreased number of sampling points with a fixed sampling frequency. In the second regime, the decrease in intensity of higher-order diffraction is steeper due to the decreased resolvable spatial frequency $\frac{1}{2P}$. In the third regime, the decrease in intensity is much steeper and it should be noted that unwanted higher order diffraction occurs at the largest diffraction orders, originating from the single supercell.

Supplementary Table 1. Simulated conditions of pixel pitch and number of pixels. To compare the intensity of diffracted beams for various pixel pitch P at the same order m , the number of pixels n is adjusted to sample signal frequency $1/\lambda$ with identical sampling frequency $1/nP$. $1/2P$ denotes the resolvable spatial frequency at the metasurface.

$\frac{1}{\lambda}$ [1/ μm]	P [nm]	n	$\frac{1}{2P}$ [1/ μm]	$\frac{1}{nP}$ [1/ μm]	m
1.58	300	12	1.67	0.278	± 5
	400	9	1.25	0.278	± 5
	500	7	1	0.286	± 5
	600	6	0.83	0.278	± 5
	700	5	0.71	0.286	± 5

Supplementary Figure 1. The intensity distribution of diffracted beams for different pixel pitches. The larger the pixel pitch, the greater the intensity decrease in higher-order diffraction. For pixel pitch of 300 nm, which satisfies the criterion derived from both the grating equation and Nyquist sampling theorem, the intensity is the most uniform.

Comment 3a:

Is the pitch of 30 μm of the supercell metasurface an optimised value? How do the numbers of the supercell pitch and the number of pixels in the metasurface supercell affect the diffraction of a flat-top target intensity and the number of diffraction spots in the imaging plane?

Our response 3a:

We thank the reviewer for the comments on the design of supercell. We hope that our response to Comment 2 has also covered the concerns about the effect of supercell to (1) intensity distribution of diffracted beams and the (4) number of diffracted beams, where the number indices follow the notation used in our response 2.

Comment 3b:

How does the depolarization effect affect the large-angle vectorial diffraction? Can the authors provide some discussion?

We thank the reviewer for the comments on the depolarization effect under large-angle condition. We found that the unintended depolarization effect was strengthened in the large-

angle vectorial diffraction by analyzing the intensity distribution of points with a viewing angle of fewer than 30° and points of more than 120°. This is because it has a curvature of the spherical wavefront as the diffraction angle gets larger. If we consider all components of electric fields including x-, y-, and z-directions, we can expect the depolarization effect and design our structural light metasurface with accurate phase modulation. We have added the discussion about depolarization effect at the manuscript and Supplementary Note 4 as below:

“Here, the depolarization effect at large-angle diffraction is also considered which distorts the beam shape into elliptical shape (Supplementary Note 4). The vectorial Debye theory can be considered to alleviate distortion, which originates from the deviation of the phase distribution between the metasurface plane and spherical wavefront¹⁰⁻¹².”

Supplementary Note 4. The depolarization effect depending on the diffraction angle.

Depolarization is a physical phenomenon in which a beam becomes depolarized under large-angle diffraction, such as when interacting with a high NA lens or wide FOV point spread metasurface because of the curvature of their spherical wavefront¹². Therefore, under large angle conditions, the point shapes are affected by the complex polarization effect because the y and z direction electric field terms are non-zero, and therefore should be considered¹³. We obtained the intensity graphs with Gaussian fitting in the x and y directions by measuring the dots at an angle less than 30 degrees from normal and points at an angle of larger than 120 degrees with a CCD from the 2D dots metasurface. The point (Supplementary Fig. 9a) at a diffraction angle less than 30 degrees shows little depolarization effect, while the intensity width in the x-direction is almost twice as long as the intensity width in the y-direction when the diffraction angle is more than 120 degrees because of the enhanced depolarization effect (Supplementary Fig. 9b).

Supplementary Figure 9. The depolarization effect depending on the diffraction angle. The larger the diffraction angle, the more prominent the depolarization phenomenon appears. **a For the point within a 30° field of view, the intensity distribution in the XY plane and the intensity plots along the x and the y directions respectively. **b** For the point outside the 120° field of view, the intensity distribution in the XY plane and the intensity plots along the x and the y directions respectively.**

Comment 4:

Based on the Fourier theorem, multiply the Fourier transform of the supercell (equivalent to a flat top across the whole imaging plane) by a 2D comb function results in an array of multi-order diffracted spots in the imaging plane (Fig. 2g). My major concern is, however, about the efficiency of designing such a small supercell metasurface as the convolution kernel in the metasurface plane. My understanding is that this small supercell needs to consider a large area of diffraction across the whole imaging plane (a few millimeters according to Fig. 3a, or even meters in the experiment), which includes large-angle diffractions. Can the authors discuss why this design approach is the best?

Our response 4:

We thank the reviewer for the concern about the efficiency of supercell and have allowed us to add more characterization about design of supercell. We hope that our response to Comment 2 has also covered the concerns about the effect of supercell to (3) diffraction efficiency, where the number indices follow the notation used in our response 2. The discussion on the advantages of our approach which includes large-angle diffractions have been covered from our response to Comment 1.

Comment 5:

I would like the authors at least compare this design approach with another intuitive case, for instance, a small metasurface supercell designed for creating a multifocal array that covers only a small projection area in the imaging plane? In this case, each metasurface supercell can be flexibly designed in terms of spot number and diffraction angle. Compared to the authors' flat top target, how about directly creating a multifocal array with a high uniformity? Like the results reported in the following papers.

[1] Opt. Lett. 36, 406-408 (2011); [2] Opt. Lett. 39, 1621-1624 (2014).

Our response 5:

We thank the reviewer for the suggestions of important papers using vectorial Debye algorithm and have allowed us to consider the depolarization effect-originated distortion at the large-angle diffraction. We have added the two important papers with discussion at the manuscript as below:

“The uniformity of diffracted beams from SLM can be resolved by calculating phase profile of SLM using vectorial Debye approximation¹⁰ which considers the depolarization effect when focusing light with a high numerical aperture objective^{11, 12}. However, the requirement of bulky objective lenses is still challenging for miniaturization of SL illuminating system.”

Reviewer #3 (Remarks to the Author):

This manuscript entitled “Metasurface-driven full-space structured light three-dimensional imaging” reports a metasurface that can generate dot array covering large field of illumination (FOI) of near 180 degrees. The dot array is center-symmetric such that the metasurface consisting of nanofins is dependent to the polarization of incident light. The authors showcase that such metasurface-generated dot array is able to performance depth sensing with a few millimeters error for a face object placed at 1 meter away from the metasurface. Lastly, a nanoimprint was shown as proof of concept for large-scale manufacturing. I highly recommend this manuscript for publication after the following itemized questions are addressed in the revision. This work is the first demonstration of metasurface-generated dot array with large FOI for depth sensing.

We appreciate the reviewer for the constructive comments on our metasurface for projecting full-space dot arrays and recommending for publication. We have carefully answered the raised comments from the reviewer as below.

Comment 1:

What is the efficiency of the metasurface shown in Fig. 2? It seems all metasurfaces have strong 0th order, what is the reason?

Our response 1:

We thank the reviewer for the comments on the important metrics of our metasurface. The overall diffraction efficiency of metasurface is 60% which is total transmitted light including zeroth-order beam normalized to incident laser intensity. Zeroth-order efficiency itself is 32%, which show discrepancies compared to simulated results (Fig. 2g). We hope that our response to Comment 2 of Reviewer 1, has also covered the reason for discrepancies with the possible solutions to alleviate the discrepancies.

Comment 2:

For the experiment results shown in Fig. 4, was the same metasurface of Fig. 2 used? I am also curious to know the spot size diameter and spot-to-spot distance. Is the metasurface's size larger than the incident laser beam diameter? What is the laser beam divergence? I ask these

questions because the diameter and spot-to-spot distance of dot array are vital parameters, which determine lateral spatial resolution. Typically, for face recognition used in mobile phones, the dot diameter needs to achieve ~ 1 mm while projecting the dot array to a screen 20 cm away from the projector.

Our response 2:

We thank reviewer for the constructive comments on the parameters that determine the lateral spatial resolution. First, we fabricated three types of metasurface, that project 2D and 1D dot arrays and 2D line arrays over full-space. The experimental results shown in Fig. 2h and Fig. 4c which project 2D dot arrays is from same metasurface. As reviewer indicated, the lateral resolution of face recognition is determined by beam diameter and spot-to-spot distance of beams. The laser we used is He-Ne laser (HNL210L, Thorlabs), which has beam divergence of 1.15 mrad, and $1/e^2$ beam diameter of 0.7 mm which is larger than size of metasurface that is 0.51 mm, therefore we cropped using pinhole to fit in metasurface. Rayleigh range and beam diameter at 20 cm away is calculated as 0.6 m and 0.735mm, respectively. Therefore, we have discussed on the spot size diameter and spot-to-spot considering only the angle cone of each diffracted beam which is determined by the number of supercells N , at the manuscript and Supplementary Figure 2 as below.

“The propagation angle of the m^{th} order diffracted beam θ_z^m with respect to the optical axis is described as $\sin^2\theta_z^m = \sin^2\theta_x^m + \sin^2\theta_y^m$, where angles about the x -, and y -axis are expressed as

$$\theta_{x,y}^m = \arcsin\left(\frac{m\lambda}{nP}\right), \quad (1)$$

where m represents the diffraction order and the largest diffraction angle $\theta_{x,y}^{m=\pm 47}$ is 82.6° . It should be noted that the uniformly sampled spatial frequency, as shown in Fig. 2g, does not ensure a uniformly spaced propagation angle, since $\sin\theta$ can be approximated to θ only at small angles. Therefore, the spacing between diffracted beams at higher orders becomes larger, as does the size of the beam. The angle cone of each diffracted beam Ω_N^m is described as $\Omega_N^m = \arcsin\left(\frac{m\lambda}{nP} + \frac{\lambda}{2NnP}\right) - \arcsin\left(\frac{m\lambda}{nP} - \frac{\lambda}{2NnP}\right)$, where N is number of repeated supercells (Supplementary Figure 2).”

Supplementary Figure 2. The effect of the number of periodic supercells on diffracted beam diameter. *a-d* Simulated far-field intensity distribution in spatial frequency domain with logarithm scale under various number of supercells. The number of supercells N is varied from $N = 2$ to $N = 6$, where every supercell consists of 10 by 10 meta-atoms on a 300 nm pitch square lattice, resulting in the same number of diffracted beams. The diameter of the diffracted beams, in other words, the angle of the cone of the diffracted beams decreases as N increases. *e* The effect of sampled spatial frequency to the diffraction angle and number of supercells to angle cone of diffracted beam. Sampling period on spatial frequency $\frac{1}{nP}$, which is function of number of meta-atoms and pixel pitch, solely determines the diffraction angle. The inset shows the m^{th} and $(m+1)^{\text{th}}$ order diffracted beams. As shown in inset, the number of supercells N is the parameter to control the angle cone of each diffracted beam, denoted as $\frac{1}{NnP}$, while not affecting the diffraction angle itself.

Comment 3:

It was mentioned through this manuscript that the metasurface has a field of illumination of 180 degrees; however, I do not see a convincing measurement data to support it. An image taken by a Fourier microscope (J.Opt.Soc.Am. A 32, 2082- 2092 (2015)) or by projecting the dot array to a hemispherical screen (Adv. Mater. 2012, 24, OP331–OP336) could be a feasible approach.

Our response 3:

We thank the reviewer for the comments on the demonstration of angle coverage from our full-space diffractive metasurfaces. The experimental results shown in Fig. 2h (Side view) shows

the projected light on the screen located at side of metasurface indicating that the light is diffracted to the large angle. We conducted two more approaches as the reviewer indicated to show the field of view more effectively, that is Fourier microscope and hemispherical screen and have added into the Supplementary Figure 6 as below.

“We also demonstrate a large FOV of our full-space diffractive metasurfaces with a hemisphere screen and Fourier microscope which allows imaging of the spatial frequency domain (Supplementary Figure 6).”

Supplementary Figure 6. Demonstration of large-angle field of view. a Schematic of the optical setup with hemisphere screen. The diameter of the hemispherical screen is 50 cm. b The 2D full-space structured light on hemisphere screen. c Fourier microscope optical setup with numerical aperture of 0.45 allowing imaging of frequency domain. The quarter of structured light is captured using CCD camera, showing diffracted beams up to 21st-order at equal intervals.

Comment 4:

Line 162 on page 6, it should read “... since $\sin(\theta)$ can be approximated to θ only at ...”.

Our response 4:

We thank the reviewer for the detailed correction. As the reviewer indicated, the sentence was modified as below.

“... $\sin(\theta)$ can be approximated to θ only at small angles.”

REVIEWERS' COMMENTS

Reviewer #1 (Remarks to the Author):

I have carefully read the revised manuscript as well as the rebuttal letter. I appreciate the authors' effort to address my comments and those of the other referees. The manuscript has been significantly improved. I believe it is an important contribution to the field of metasurface and 3D imaging, and recommend its publication in Nature Communications.

Reviewer #2 (Remarks to the Author):

I am satisfied with the authors' responses and the revised manuscript, which can be accepted for publication in my view.

Reviewer #3 (Remarks to the Author):

In the revision, the authors have replied my questions with solid and rigorous data. I definitely support the publication of this manuscript in its current form.

Reviewer #1 (Remarks to the Author):

I have carefully read the revised manuscript as well as the rebuttal letter. I appreciate the authors' effort to address my comments and those of the other referees. The manuscript has been significantly improved. I believe it is an important contribution to the field of metasurface and 3D imaging, and recommend its publication in Nature Communications.

We are pleased to see the positive comment and final recommendation. Thanks to the careful comments and useful suggestions, we can significantly improve the quality of our work. Thank you again for your careful review.

Reviewer #2 (Remarks to the Author):

I am satisfied with the authors' responses and the revised manuscript, which can be accepted for publication in my view.

We thank the reviewer for constructive comments on our work which could significantly improve the quality of our work. Thanks for the positive comment and final recommendation. We are hoping that the proposed work could contribute to the field of metasurface for 3D depth imaging.

Reviewer #3 (Remarks to the Author):

In the revision, the authors have replied my questions with solid and rigorous data. I definitely support the publication of this manuscript in its current form.

We feel very thankful for the reviewer's positive comment and the final recommendation. We are delighted that the quality of our work could be improved thanks to the careful and constructive suggestions.